# CoFe Alloy-Coupled Mo_2_C Wrapped by Nitrogen-Doped Carbon as Highly Active Electrocatalysts for Oxygen Reduction/Evolution Reactions

**DOI:** 10.3390/nano13030543

**Published:** 2023-01-29

**Authors:** Jiahao Xie, Yu Miao, Bin Liu, Siliang Shao, Xu Zhang, Zhiyao Sun, Xiaoqin Xu, Yuan Yao, Chaoyue Hu, Jinlong Zou

**Affiliations:** 1Key Laboratory of Functional Inorganic Material Chemistry, Ministry of Education of the People’s Republic of China, School of Chemistry and Materials Science, Heilongjiang University, Harbin 150080, China; 2MIIT Key Laboratory of Critical Materials Technology for New Energy Conversion and Storage, School of Chemistry and Chemical Engineering, Harbin Institute of Technology, Harbin 150080, China

**Keywords:** bifunctional electrocatalysts, in-situ X-ray diffraction, Prussian blue analogues, synergistic effect, transition metal carbides

## Abstract

Molybdenum carbide (Mo_2_C) with a Pt-like d-band electron structure exhibits certain activities for oxygen reduction and evolution reactions (ORR/OER) in alkaline solutions, but it is questioned due to its poor OER stability. Combining Mo_2_C with transition metals alloy is a feasible way to stabilize its electrochemical activity. Herein, CoFe-Prussian blue analogues are used as a precursor to compound with graphitic carbon nitride and Mo^6+^ to synthesize FeCo alloy and Mo_2_C co-encapsulated N-doped carbon (NG-CoFe/Mo_2_C). The morphology of NG-CoFe/Mo_2_C (800 °C) shows that CoFe/Mo_2_C heterojunctions are well wrapped by N-doped graphitic carbon. Carbon coating not only inhibits growth and agglomeration of Mo_2_C/CoFe, but also enhances corrosion resistance of NG-CoFe/Mo_2_C. NG-CoFe/Mo_2_C (800 °C) exhibits an excellent half-wave potential (E_1/2_ = 0.880 V) for ORR. It also obtains a lower OER overpotential (325 mV) than RuO_2_ due to the formation of active species (CoOOH/β-FeOOH, as indicated by in-situ X-ray diffraction tests). E_1/2_ shifts only 6 mV after 5000 ORR cycles, while overpotential for OER increases only 19 mV after 1000 cycles. ORR/OER performances of NG-CoFe/Mo_2_C (800 °C) are close to or better than those of many recently reported catalysts. It provides an interfacial engineering strategy to enhance the intrinsic activity and stability of carbides modified by transition-metals alloy for oxygen electrocatalysis.

## 1. Introduction

The development of renewable energy conversion and storage systems is urgent and necessary in the current society where fossil fuels are in short supply and the environment is severely damaged [1,2]. The electrochemical oxygen reduction and evolution reactions (ORR/OER) are the key half-reactions on the air cathode of energy conversion systems, including metal-air batteries (MABs), direct methanol fuel cells, and hydrogen fuel cells [2,3]. However, the slow kinetics of these two half-reactions limit their large-scale developments for practical application in MABs [1,4]. Currently, Pt-based materials usually exhibit impressive ORR activity, while IrO_2_/RuO_2_ materials are the main catalysts for OER [5,6]. Unfortunately, the practical use of these noble metal-based catalysts for ORR/OER is hindered by their high cost and low catalytic stability with a high susceptibility to OH^−^ ion poisoning [7,8]. Therefore, the development of cheap, efficient, and stable ORR/OER catalysts for MABs is of great significance for environmental protection and economic development.

Carbon materials doped with non-precious transition metal carbides have proven to be good alternatives to noble metal catalysts by virtue of their low cost and excellent ORR/OER dual functionality [9]. Among them, molybdenum carbide has received a lot of attention as an active material due to its excellent electrical conductivity, good corrosion resistance, and d-band electronic structure similar to that of Pt group metals [6,10]. For example, Du et al. synthesize the highly stable molybdenum carbide (Mo_2_C) nanodots anchored within the 3D carbon nanocages (denoted as Mo_2_C/C) by pyrolysis of molybdenum tartrate dry gels at high temperatures. Mo_2_C/C-0.5 requires only a low overpotential of 164 mV (η_10_) for HER in 1 M KOH solution [11]. Guo et al. develop a simple method for loading ultrafine Mo_2_C nanoparticles onto N-doped carbon nanosheets to obtain mesoporous 2D nanosheets (MMo_2_C/NCS) with a unique mesoporous nature, which shows an excellent ORR catalytic activity (half-wave potential (E_1/2_) of 0.85 V) [12]. However, the high-temperature calcination not only causes the agglomeration of small particles during the synthesis process, but it also generates the corresponding metallic impurities, which may lead to the formation of large-sized Mo_2_C particles and reduce the number of active sites, ultimately lowering the electrocatalytic performance [13].

As reported recently, controlled doping of transition metals (Fe, Co, Ni, etc.) or their alloys with promising electrochemical properties into a nanomaterial can effectively increase its electrical conductivity and catalytic activity [1,8,14]. Kou et al. find that the heterostructures composed of Co-species and Mo_2_C together have better OER catalytic properties than those of single Co-based or Mo_2_C material [15]. The OER overpotential of Mo_2_C catalyst alone is 310 mV, which changes to 190 mV at 10 mA cm^−2^ after the addition of Co species. The excellent OER performance is attributed to the chemical coupling that occurs at the interface between Co-species and Mo_2_C, which enhances the electrophilicity and OH^−^ affinity and forms a hydrated oxide layer on the surface of Co-species [8,15]. Gu et al. synthesize non-precious metal electrocatalysts consisting of Mo_2_C nanosheets arranged vertically on the surface of Co@NC. Thanks to the interfacial effect between Co@NC and Mo_2_C, Mo_2_C/Co@NC exhibits excellent tri-functional electrochemical properties of ORR (E_1/2_ = 0.86 V), OER (η_10_ = 308 mV), and HER (η_10_ = 51 mV) [1]. Peng et al. report the in-situ synthesis of a Mo_2_C/FeNi alloy-combined catalyst with high electrical conductivity (FeNi−Mo_2_C/carbon flower) by using a hydrothermal method [10]. The unique flower-like porous structure can provide efficient mass transfer pathways to obtain the excellent electrocatalytic performance [10]. Moreover, due to the synergistic effect between geometric design and electronic modification, more active sites on the flower-like structure of the FeNi-Mo_2_C/carbon catalyst are exposed, which correspondingly accelerates the electron transfer to enhance the OER activity (η_10_ = 228 mV) [10]. These studies clearly show that the electrochemical performance of Mo_2_C doped with metallic species or alloys is much better than that of the undoped one. Therefore, it is reasonable to combine the Mo_2_C with the transition metal alloy to enhance electrochemical performance [15]. However, the ORR/OER dual functions jointly promoted by the composite with Mo_2_C and transition metal alloys are still not comprehensive and need to be further investigated.

Herein, we use solid nanospheres (CoFe-Prussian blue analogues/polyvinyl pyrrolidone, CoFe-PBA/PVP), Mo^6+^, and graphitic carbon nitride (g-C_3_N_4_) to prepare a non-precious metal ORR/OER bifunctional electrocatalyst (NG-CoFe/Mo_2_C) by using a simple carbonization method. The catalyst consists of an outer graphitized carbon layer and an inner metal species (CoFe/Mo_2_C). The special structure not only provides abundant electron transfer channels and facilitates the exposure of more active sites, but it also protects the inner metal species from alkaline corrosion to enhance the catalytic stability. The close contact between Mo_2_C and CoFe alloy can generate a large number of active sites for ORR/OER. Moreover, N species (originated from g-C_3_N_4_) and Mo_2_C can jointly act as electron donors for the C atoms in the shell. It is expected that the NG-CoFe/Mo_2_C catalysts should exhibit better catalytic activity and stability than those of commercial Pt/C (ORR) and RuO_2_ (OER) in alkaline electrolytes, providing a new perspective on the development of non-precious metal carbide/alloy-based bifunctional electrocatalysts. 

## 2. Materials and Methods

### 2.1. Synthesis of CoFe-PBA/PVP and g-C_3_N_4_ Precursors

The CoFe-PBA/PVP hybrid precursor was synthesized by using a co-precipitation method [3,16]. 2.646 g (9 mM) of sodium citrate (Na_3_C_6_H_5_O_7_∙2H_2_O), 1.746 g (6 mM) of cobalt nitrate hexahydrate (Co(NO_3_)_2_∙6H_2_O), and 1.2 g of PVP were dissolved in 100 mL of anhydrous ethanol to form a clarified solution (Solution A). A total of 1.317 g of potassium hexacyanoferrate (III) (K_3_[Fe(CN)_6_]) was dissolved in 40 mL of H_2_O to form Solution B, which was then added dropwise to Solution A under magnetic stirring for 5 min. The above mixed solution was aged at room temperature (25 °C) for 2 h. The precipitate was collected by centrifugation and dried at 60 °C for 12 h. For synthesis of g-C_3_N_4_ nanotubes, 5 g of melamine was put into a crucible and heated at 550 °C for 4 h with a heating rate of 5 °C min^−1^ in a muffle furnace. After cooling naturally to room temperature (25 °C), a yellow solid (g-C_3_N_4_ nanotubes) was obtained and ground into powder for future use [17,18].

### 2.2. Synthesis of Fe-N/C Catalyst

An amount equal to 0.2 g of the above synthesized Co-Fe PBA/PVP precursor, 0.2 g of (NH_4_)_6_Mo_7_O_24_·4H_2_O, and 0.2 g of g-C_3_N_4_ were dispersed in 20 mL of H_2_O, and then the mixture was aged for 6 h at room temperature (25 °C). The dark green precipitate was collected by centrifugation and dried at 60 °C for 12 h (denoted as CoFeMo-CN). As reported previously, Mo^6+^ ((NH_4_)_6_Mo_7_O_24_·4H_2_O) was coupled to the CoFe-PBA/PVP precursor by coordination between the pyridine group of PVP and Mo^6+^ to form the solid spheres loaded with a large number of Mo^6+^ on the surface [14]. The CoFeMo-CN sample was annealed at 700, 750, 800, 850, or 900 °C under N_2_ atmosphere for 2 h at a heating rate of 3 °C min^−1^ to obtain the NG-CoFe/Mo_2_C-x (x = 700, 750, 800, 850, or 900) catalyst. The synthesis process (Figure 1) had three steps, including template growth, surface attachment, and carbonization. For comparison, the catalyst without (NH_4_)_6_Mo_7_O_24_·4H_2_O was also prepared using the same procedure and denoted as NG-CoFe. Characterization methods for the prepared materials including X-ray diffraction (XRD), Raman spectroscopy, contact angle, X-ray photoelectron spectroscopy (XPS), Fourier Transform Infrared (FTIR) spectroscopy, scanning electron microscopy (SEM), transmission electron microscopy (TEM), energy dispersive x-ray spectroscopy (EDS), and N_2_ adsorption/desorption isotherms were described in the ‘Appendix A’.

### 2.3. Electrochemical Measurements

Electrochemical ORR/OER measurements were performed in a three-electrode configuration with a CHI 760 E electrochemical workstation, using a glassy carbon electrode as the working electrode. A homogeneous ink was prepared by mixing 0.005 g of the synthesized electrocatalyst, 50 µL of nafion, and 100 µL of anhydrous ethanol together under ultrasonication for 0.5 h. The resulting catalyst ink was dropped onto the glassy carbon electrode with an electrocatalyst loading of approximately 0.2 mg cm^−2^. In this study, commercial Pt/C (10 wt.%) and RuO_2_ were used as reference catalysts for ORR and OER, respectively. Among them, commercial Pt/C (10 wt.%, Shanghai Hosen Electric Co., Ltd., Shanghai, China) used VXC72R conductive carbon black as the carbon carrier. Electrochemical tests including cyclic voltammetry (CV), linear scanning voltammetry (LSV), electrochemical impedance spectroscopy (EIS), rotating disk electrode (RDE), rotating ring disk electrode (RRDE), double-layer capacitance (Cdl), electrochemical active surface area (ECSA), transformation frequency (TOF), Faraday efficiency, chronoamperometry (CA), and accelerated durability testing (ADT) were described in the ‘Appendix A’.

## 3. Results

### 3.1. Structural and Compositional Analyses

To determine the crystal structure of the NG-CoFe/Mo_2_C-x catalysts, powder XRD tests are performed. As shown in Figure 1a, the peaks (NG-CoFe and NG-CoFe/Mo_2_C-800) at around 44.9° and 65.4° exactly match the (110) and (200) planes of the CoFe alloy (JCPDS: 49-1568), respectively [19]. The diffraction peaks at 2θ = 34.3°, 38.0°, 39.4°, 52.1°, 61.5°, 69.7°, 72.4°, 74.7°, and 75.5° can be indexed to the (100), (022), (101), (102), (110), (103), (200), (112), and (201) crystalline planes of Mo_2_C (JCPDS:35-0787), respectively. It can be observed that all the diffraction peaks of NG-CoFe/Mo_2_C-800 match well with those of the CoFe alloy and Mo_2_C. The synergies between CoFe alloy and Mo_2_C should enhance the electrocatalytic activity of the NG-CoFe/Mo_2_C-x catalyst. Specifically, the CoFe alloy can provide a large number of electrons to fill the antibonding orbitals of Mo, which can weaken the O−O bond and further promote the desorption of O-intermediates [20]. Figure 1b exhibits the XRD patterns of the catalysts carbonized at different temperatures. It can be noticed that the diffraction peaks of these catalysts are basically the same as those of NG-CoFe/Mo_2_C-800, revealing that the NG-CoFe/Mo_2_C-x catalysts are successfully synthesized. In addition, the XRD patterns of NG-CoFe/Mo_2_C-x (x = 700, 750, 800, 850, and 900) show the broad diffraction peaks (unobvious) at around 24.3°, corresponding to the (002) plane of the graphitized carbon.

To analyze the carbon structure defects of catalysts, Raman tests are performed for NG-CoFe/Mo_2_C-x (x = 700, 750, 800, 850, and 900). As shown in Figure 2a, both NG-CoFe/Mo_2_C-800 and NG-CoFe materials have two strong peaks at around 1352 and 1578 cm^−1^, which can be attributed to the D and G bands, respectively. Generally, the graphitization degree of carbon materials is positively correlated with the I_D_/I_G_ ratio [21]. The I_D_/I_G_ ratio of NG-CoFe/Mo_2_C-800 is 0.573, indicating that its carbon skeleton has many defects and a high degree of graphitization. In contrast, the graphitization degree (I_D_/I_G_ ratio of 0.901) of NG-CoFe (without Mo_2_C) is higher than that of NG-CoFe/Mo_2_C-800. It indicates that the incorporation of the Mo-species generates a large number of defects in the carbon lattice, which may significantly enhance the ORR activity [22]. The changes in the surface groups of NG-CoFe/Mo_2_C-800 before and after carbonization are further analyzed by FT-IR spectroscopy (Figure 2b). For the pre-fired CoFeMo-CN sample, the peaks at around 589, 1572, 1635, and 2085 cm^−1^ are caused by the stretching vibrations of Fe−C bond, the bending vibration of C−C bond, the resonance vibration of C=C bond, and the vibration of −C≡N− bond, respectively [23,24]. In addition, the NG-CoFe/Mo_2_C-800 catalyst mainly has two carbon characteristic vibrations at around 1556 and 2085 cm^−1^, which correspond to the stretching vibrations of C−N and −C≡N−, respectively [25]. According to the comparison, it can be found that the −C≡N− bond continues to exist before and after the calcination, while the C−N bond is newly generated, indicating that some of the −C≡N− bonds may be broken and converted into C−N bonds at high temperatures [25].

The electrocatalytic properties are closely related to the valence electronic states of elements on the surface of the catalyst [26]. As shown in the XPS survey spectra (Figure 3a), all of the NG-CoFe/Mo_2_C-x catalysts consist of six elements including C, N, O, Fe, Co, and Mo. Among them, the existence of oxygen species may be ascribed to the introduction of a small amount of air during the carbonization process. The percentage contents (wt.%) of C 1s, N 1s, O 1s, Co 2p, Fe 2p, and Mo 3d in NG-CoFe/Mo_2_C-x are shown in Appendix A. The C 1s peak of NG-CoFe/Mo_2_C-800 can be deconvoluted into two peaks of 284.6 eV (C−C) and 285.5 eV (C−N) (Figure 3b). The presence of C−N bond can also indicate the successful doping of N atoms into the carbon skeleton [14]. Figure 3c shows the N 1s spectrum of NG-CoFe/Mo_2_C-800, which can be divided into three peaks including 398.4 (pyridine N), 399.8 (pyrrole N) and 401.1 eV (graphitic N). It is expected that the N species in the catalyst can enhance the ORR activity. Generally, pyrrole N can accelerate the breaking of the O−O bond and the oxygen trapping [27]. Pyridine N can provide a large number of defects and increase the spin density of C atoms, while graphitic N facilitates charge transfer to enhance ORR activity [27].

As shown in Figure 3d, the high-resolution spectrum of Fe 2p can be deconvoluted into six peaks of 707.4, 720.1, 710.9, 724.8, 714.4, and 732.5 eV. The peaks at 707.4 and 720.1 eV correspond to Fe^0^, while the peaks at 710.9 and 724.8 eV correspond to Fe^2+^ [28]. The Fe 2p band located at around 714.4 and 732.5 eV are attributed to the satellite peaks. The presence of Fe^0^ indicates the successful synthesis of FeCo alloys under high-temperature pyrolysis conditions, while the infiltration of oxygen during carbonization is responsible for the appearance of Fe^2+^ (the oxidation of Fe^0^ by O_2_) [28]. In Figure 3e, the high-resolution spectrum of Co 2p can be divided into six peaks. The binding energies of 778.2 and 793.5 eV correspond to Co^0^ of the 2p_3/2_ and 2p_1/2_ orbitals, respectively, while the peaks at around 781.4 and 796.9 eV are assigned to the 2p_3/2_ and 2p_1/2_ spin orbitals of Co^2+^ [29]. In addition, the peaks around 787.1 and 801.1 eV are satellite peaks, further revealing the formation of CoFe alloys in the composite [27,30,31]. The change in binding energy may be due to the establishment of a new equilibrium of the charge distribution between CoFe and Mo_2_C. In Figure 3f, the high-resolution spectrum of Mo 3d is divided into two pairs of peaks with binding energies of 227.8/231.5 and 232.1/235.1 eV, corresponding to Mo^2+^ and Mo^6+^, respectively [32]. The presence of Mo^6+^ is attributed to the surface oxidation of Mo_2_C, which is consistent with previous study [33]. The XPS results demonstrate the successful synthesis of the complex structure (Mo_2_C coupled with FeCo alloy). The high-resolution XPS spectra of NG-CoFe/Mo_2_C-x (x = 700, 750, 850, and 900) catalysts with C 1s, N 1s, Fe 2p, Co 2p, and Mo 3d are shown in Appendix A.

As shown in Figure 4a, the FESEM image of the CoFe-PBA/PVP hybrid precursor shows a spherical structure with a particle diameter of approximately 20 nm. The PVP layer covering the surface can effectively prevent the agglomeration of CoFe-PBA. After doping with Mo^6+^ and g-C_3_N_4_ (Figure 4b), the regular solid spheres become irregular aggregates. After carbonization, the small nanoparticles are stacked together to form a complex structure with a high level of porosity (Figure 4c). The morphology change during the carbonization process may be due to the effect of surface tension generated by the volatilization of residual solvents [30]. As shown in Figure 4d–g, the morphologies of NG-CoFe/Mo_2_C-x show the bamboo-like nanotube structures after carbonization at different temperatures. The stacked pores (porous structure) of NG-CoFe/Mo_2_C-x can increase the number of catalytic active sites and mass transfer channels, which should correspondingly enhance the ORR/OER performance [34]. Furthermore, as shown in Figure 4h–n, EDS element mapping images show the presence of five elements including C, N, Fe, Co, and Mo on the surface of NG-CoFe/Mo_2_C-800, demonstrating the successful combination of the CoFe alloy and Mo_2_C.

The nanostructure of NG-CoFe/Mo_2_C-800 is further evaluated using TEM images. Many sub-spherical structured particles marked with red circles can be seen in Figure 5a. Most of the Mo_2_C particles and CoFe alloys are encapsulated in a robust carbon shell (Figure 5b). Figure 5c clearly shows the lattice stripes with spacing of 0.18, 0.20, and 0.34 nm, which correspond to the (102) crystal plane of Mo_2_C, the (110) crystal plane of CoFe alloy, and the (002) crystal plane of graphitized carbon, respectively. It proves the successful synthesis of Mo_2_C and CoFe alloy embedded in the carbon skeleton. The presence of graphitized carbon not only effectively facilitates the charge transfer among Mo_2_C, CoFe alloy, and carbon during electrocatalysis, but it also reduces the corrosion of Mo_2_C and CoFe alloy by the electrolyte and improves the stability of the catalyst [6]. Moreover, the close contact of CoFe alloys and Mo_2_C nanoparticles can improve the electron transfer from the CoFe/Mo_2_C nanoparticles to the carbon matrix, which in turn enhances the catalytic performance [35]. As shown in Figure 5d, the selected area electron diffraction (SAED) also reveals the successful synthesis of Mo_2_C and FeCo alloy in the graphitized carbon structure, consistent with the HRTEM results.

N_2_ adsorption–desorption isotherms and pore size distribution curves for CoFePBA/PVP, NG-CoFe, and NG-CoFe/Mo_2_C-800 are displayed in Figure 6a–c. The specific surface areas (SSAs) of NG-CoFe/Mo_2_C-800 are 322.03 m^2^ g^−1^, which is much higher than those of CoFePBA/PVP (145.74 m^2^ g^−1^) and NG-CoFe (178.57 m^2^ g^−1^), respectively (Appendix A). The N_2_ adsorption–desorption isotherms of these catalysts reveal the typical type IV isotherms with a (mainly) mesoporous characteristic. It reveals that the carbon skeleton is highly porous, which should facilitate the mass (oxygen-related substances) diffusion via the internal pore channels and the exposure of abundant active sites, thus accelerating the ORR/OER kinetics. As indicated previously, the metal cores and porous carbon structure can together enhance the electrocatalytic activity [28]. In order to study the wettability of the catalysts, contact angle measurements are performed. Figure 6d–f show that the contact angles of CoFePBA/PVP, NG-CoFe, and NG-CoFe/Mo_2_C-800 are 15.46°, 13.65°, and 8.96°, respectively. The hydrophilic property of NG-CoFe/Mo2C-800 should be promoted by embedding the hydrophilic CoFe alloy/surface oxides in the carbon skeleton. Moreover, the large surface area with abundant mesopores is also responsible for the better wettability. It indicates that the NG-CoFe/Mo_2_C-800 catalyst with a hydrophilic surface facilitates full contact with the electrolyte. It is reported that high wettability can theoretically improve the adsorption of reactants or products on the catalytic interface, thus improving the electrocatalytic activity [36]. 

### 3.2. Electrocatalytic Activities of NG-CoFe/Mo_2_C-x Catalysts for ORR

The ORR activities of NG-CoFe/Mo_2_C-x are evaluated using CV tests in oxygen-saturated 0.1 M KOH electrolyte [37]. In Figure 7a, NG-CoFe/Mo_2_C-800 has an obvious reduction peak at 0.880 V, which is higher than those of commercial 10 wt.% Pt/C (0.830 V) and NG-CoFe (0.802 V). It suggests that the excellent ORR activity is mainly due to the synergies among Mo_2_C and CoFe, which can promote the capture of the first electron and accelerate the cleavage of the O=O bond, leading to a significant reduction of the activation potential of the adsorbed O_2_, thus enhancing the ORR activity [38,39,40]. As reported previously, Mo_2_C with a strong negative electronic property has a similar d-band structure to Pt-group metals, which makes it have promising catalytic properties for ORR [37,38]. N dopants (especially pyridine-N) can form the Lewis bases on the adjacent carbon to accelerate the adsorption of oxygen molecules to further enhance the ORR efficiency [12]. In Figure 7b, the reduction peaks of NG-CoFe/Mo_2_C-x (x = 700, 750, 850, and 900) appear at 0.866, 0.856, 0.845, and 0.831, respectively. It reveals that the NG-CoFe/Mo_2_C-800 catalyst has the highest ORR catalytic activity (0.880 V). To gain deeper insight into the ORR dynamics, LSV tests of NG-CoFe/Mo_2_C-x catalysts are conducted in an O_2_-saturated 0.1 M KOH solution at 1600 rpm. In Figure 7c, NG-CoFe/Mo_2_C-800 exhibits the highest ORR activity with a promising E_1/2_ of 0.880 V, which is 50 and 78 mV higher than those of Pt/C (0.830 V) and NG-CoFe (0.802 V), respectively. In Figure 7d, the activity of NG-CoFe/Mo_2_C-800 is superior to other as-prepared catalysts, indicating that more available active sites are exposed on NG-CoFe/Mo_2_C-800 to accelerate the ORR process [41,42]. As reported by our group, the surface Co^2+^ species with neighboring oxygen vacancy can be considered as the active sites, which can promote the adsorption and activation of O_2_ for ORR [35,36]. Moreover, Co^2+^ can also transfer electrons to the O_2_ absorbed on the active site, thus weakening and breaking the O−O bond to facilitate the ORR process on the surface of NG-CoFe/Mo_2_C-800 [35,36].

As shown in Figure 8a, the kinetic current density (J_k_) of the NG-CoFe/Mo_2_C-800 catalyst is 3.1 mA cm^−2^ at 0.82 V, which is 2.82 times higher than that of NG-CoFe (1.1 mA cm^−2^) and 1.19 times higher than that of commercial Pt/C (2.6 mA cm^−2^). To summarize, NG-CoFe/Mo_2_C-800 catalyst has the highest kinetic current density, further illustrating its favorable catalytic performance [43]. The lower Tafel slope usually confirms the more promising ORR kinetics of the catalyst [44]. Figure 8b shows the Tafel plots of NG-CoFe/Mo_2_C-800 and Pt/C. The Tafel slope of NG-CoFe/Mo_2_C-800 (92.47 mV dec^−1^) is better than that of commercial Pt/C (241.16 mV dec^−1^), indicating that NG-CoFe/Mo_2_C-800 catalyst has the faster reaction kinetics. By comparing with NG-CoFe, the superior catalytic performance of NG-CoFe/Mo_2_C-x should be ascribed to the incorporation of Mo_2_C with the electrocatalytic synergies among CoFe alloy, N-doped carbon, and Mo_2_C [45,46]. As shown in Figure 8c, the charge transfer resistance (R_CT_) values of NG-CoFe/Mo_2_C-800, NG-CoFe, and Pt/C are 17.73, 56.92, and 64.12 Ω, respectively. Moreover, the R_CT_ values of NG-CoFe/Mo_2_C-x (x = 700, 750, 850, and 900) are 34.01, 42.98, 18.65, and 28.26 Ω, respectively (Appendix A). The lower the R_CT_ and the faster the charge transfer rate of the catalyst [47]. It clearly shows that NG-CoFe/Mo_2_C-800 has a faster electron transfer kinetics (Appendix A). These results verify that the introduction of Mo_2_C can improve the electronic conductivity to thus promote the charge transfer ability of NG-CoFe/Mo_2_C-800 [47]. As deduced, the smooth charge transfer between metal species (Mo_2_C and CoFe) with high surface activities and a carbon skeleton (electron-accepting) can be partly attributed to the electron-donating effects of Mo_2_C and CoFe, which eagerly induces a charge-transfer from Mo_2_C/CoFe to carbon.

The ORR kinetics of NG-CoFe/Mo_2_C-800 and Pt/C are investigated at different rotational rates (Figure 9a,b). The corresponding Koutecky-Levich (K-L) plots from LSV curves are shown in the insets of Figure 9a,b, and the average number of transferred electrons (n) is calculated based on the slope of the K-L plot (Appendix A). The K-L plots of NG-CoFe/Mo_2_C-800 and Pt/C nearly overlap in the potential range of 0.3 to 0.6 V. All curves show excellent linear parallelism with n close to the theoretical value of 4.0, indicating that the catalyst facilitates the 4e- reduction pathway for oxygen [48]. As shown in Figure 9c,d, the H_2_O_2_ yields and the number of transferred electrons (n) in alkaline electrolytes can be calculated by using the RRDE tests. The H_2_O_2_ yields of NG-CoFe/Mo_2_C-800, NG-CoFe, and Pt/C are between 10.4 and 13.4%, indicating that the NG-CoFe/Mo_2_C-800 catalyst has an efficient selectivity for the generation of H_2_O, further confirming the 4e^−^ ORR process (mainly) [49]. As shown in Figure 9e, the stability tests are performed in an O_2_-saturated 0.1 M KOH solution for 30,000 s of continuous testing. The retention of the current density of the NG-CoFe/Mo_2_C-800 catalyst (84.74%) is higher than that of Pt/C (58.30%), and the loss of current density can be partly attributed to the oxidation of metal species during the electrocatalytic process [50]. It is obvious that the NG-CoFe/Mo_2_C-800 catalyst has better stability, which may be due to the special structure with CoFe and Mo_2_C encapsulated (protected) by the N-doped carbon [51].

As shown in Figure 9f, the current density of Pt/C changes drastically at 300 s when 3 mL of methanol is added to the electrolyte for the CA test, while the curve of the NG-CoFe/Mo_2_C-800 catalyst does not show any fluctuation. The excellent methanol resistance of NG-CoFe/Mo_2_C-800 is probably due to the large surface area of the carbon matrix, where OH^−^ can be effectively adsorbed to enhance the methanol tolerance during ORR [26]. In addition, the N-doped carbon matrix allows for the particle-wrapped configuration and helps to prevent the aggregation of Mo_2_C and CoFe nanoparticles, which in turn enhances the catalytic and structural stabilities of the catalyst [12]. In the accelerated durability tests (ADTs) (Appendix A), the E_1/2_ of NG-CoFe/Mo_2_C-800 has a slight shift of 6 mV after 5000 cycles, which is lower than that of Pt/C (15 mV). It indicates that NG-CoFe/Mo_2_C-800 can be considered as an excellent ORR electrocatalyst with high catalytic stability and methanol resistance. These promising properties of NG-CoFe/Mo_2_C-800 are mainly attributed to its special structure and components, which not only provide sufficient active sites by coupling N-doped carbon with Mo_2_C and CoFe nanoparticles, but also provide multiple pathways for the efficient transport of oxygen molecules and electrolytes [10].

### 3.3. Electrocatalytic Activities of NG-CoFe/Mo_2_C-x Catalysts for OER

Figure 10a shows that NG-CoFe/Mo_2_C-800 can achieve a current density of 10 mA cm^−2^ at an overpotential of 325 mV for OER, which is much smaller (better) than those of NG-CoFe (422 mV) and RuO_2_ (389 mV). The OER overpotentials (E_j_ = 10) for NG-CoFe/Mo_2_C-x (x = 700, 750, 800, 850, and 900) catalysts are 351, 341, 343, and 339 mV (Appendix A), respectively. It also indicates that CoFe alloy alone provides a limited OER activity for the NG-CoFe catalyst, while the catalyst with Mo_2_C provides a promising OER performance [52,53,54]. It is reported that the strong interactions between the two elements (Co and Fe) can effectively improve the electrocatalytic activity of alloys for OER [55,56]. The close contact between Mo_2_C and CoFe (heterojunction) should lead to the modification of the structure of CoFe alloy, contributing to the formation of highly active CoOOH/FeOOH [57,58]. Moreover, although the protecting role of NG can delay the oxidation of metallic species, the formation of ionic states (Co^2+^/Fe^2+^) of Co/Fe is inevitable, which can effectively accelerate the formation of the key active species (CoOOH/FeOOH) for OER [59]. The catalytic activity is mainly due to the synergistic effect of the heterogeneous structure (CoOOH/FeOOH) formed by the reaction at the three-phase interface [60]. Specifically, the interaction between CoFe and Mo_2_C also accelerates the electron transfer rate, enhances the electrophilic effect of Co ions and Fe ions, and strengthens the adsorption capacity for OH^−^, hence increasing the OER activity [60]. To further investigate the OER performance of the catalysts, the corresponding Tafel slopes are obtained from the LSV curves. The Tafel slope is the size of the overpotential required for a 10-fold increase in the exchange current, thus, the smaller the slope, the faster the OER dynamics [52]. In Figure 10b, the Tafel slope of NG-CoFe/Mo_2_C-800 (75.49 mV dec^−1^) is much smaller than that of RuO_2_ (173.03 mV dec^−1^). The Tafel slopes of NG-CoFe/Mo_2_C-x (x = 700, 750, 850, and 900) are 76.39, 86.29, 76.39, and 74.87 mV dec^−1^, respectively. It shows that NG-CoFe/Mo_2_C-800 has a better OER catalytic kinetics [61,62]. In Figure 10c (EIS curve), NG-CoFe/Mo_2_C-800 has a smaller R_CT_ (18.33 Ω) than those of NG-CoFe (46.75 Ω) and RuO_2_ (25.94 Ω). The active species (CoOOH/FeOOH)-wrapped configuration of NG-CoFe/Mo_2_C-800 may play important roles in enhancing the charge transfer during OER [63,64,65]. In addition, the N-doped carbon skeleton substantially improves the electrical conductivity, provides high surface area to contact with electrolytes, and disperses the generated CoOOH/FeOOH species without aggregation [66,67].

It is well known that the ECSA can be used to indicate the number of active sites exposed by the catalyst and also to reflect OER catalytic activity [68,69]. ECSA is evaluated using cyclic voltammetry based on CV curves at different scan rates to calculate the double-layer capacitance (C_dl_) values (Figure 10d and Appendix A). As shown in Figure 10d, the C_dl_ value of NG-CoFe/Mo_2_C-800 (17.45 mF cm^−2^) is the larger than those of NG-CoFe (1.67 mF cm^−2^) and RuO_2_ (1.40 mF cm^−2^). The ECSAs of NG-CoFe/Mo_2_C-800, NG-CoFe, and RuO_2_ are 436.25, 41.75, and 35.00 mF, respectively. In agreement with the LSV (OER) results, the ECSA of NG-CoFe/Mo_2_C-800 is larger than those of NG-CoFe and RuO_2_. In Figure 10e, the electron transfer number (n) is calculated using RRDE, and n is approximately 4.0 for NG-CoFe/Mo_2_C-800. It shows that the dominant OER pathway is the 4e^−^ transfer process (4OH^−^ → 2H_2_O + O_2_ + 4e^−^) [67,70]. As shown in Appendix A, the turnover frequency (TOF) value calculated for NG-CoFe/Mo_2_C-800 (0.790 s^−1^) at 10 mA cm^−2^ is superior to that of RuO_2_ (0.354 s^−1^). As shown in Figure 10f, the Faraday efficiency (93.2%) of NG-CoFe/Mo_2_C-800 is obtained by performing the RRDE test, further validating the 4e^−^ pathway for OER and also demonstrating that NG-CoFe/Mo_2_C-800 has an excellent capability for energy conversion [71,72].

Previous studies have mentioned that Mo_2_C can significantly improve the OER performance of the catalyst, but it is questioned due to its poor stability [20,54]. Specifically, Mo is easily oxidated to Mo-based oxides during OER, resulting in its poor stability [20]. To investigate the OER stability of NG-CoFe/Mo_2_C-800, continuous LSV measurements are performed (Figure 11a). The overpotential in the LSV curve increases by only 19 mV compared to the initial one after 1000 cycles, and finally the current density is retained by 91.8% (inset of Figure 11a). In Figure 11b, the crystalline phase change of NG-CoFe/Mo_2_C-800 is tested during OER by using the in situ XRD technique. The diffraction peaks at around 20.24° and 50.58° correspond to the (0 0 3) and (0 1 5) lattice of CoOOH (JCPDS No. 07-0169), respectively [73]. In addition, the diffraction peaks at around 26.70° and 55.90° correspond to the (3 1 0) and (5 2 1) lattices of β-FeOOH (JCPDS No. 34-1266), respectively [63]. These new active components (CoOOH/FeOOH) should be produced by the oxidation of the Co/Fe species during OER. As reported previously, CoOOH and FeOOH are the main active substances for OER [71,73]. Usually, the value of overvoltage (ΔE = E_j=10_−E_1/2_) is used to evaluate efficiency loss, which can validly indicate the bifunctional electrocatalytic activity of the catalyst [49]. A smaller ΔE value indicates a lower efficiency loss and higher catalytic activity as a reversible oxygen electrode [67,74]. The as-prepared NG-CoFe/Mo_2_C-800 catalyst exhibits excellent catalytic activity with a ΔE of 0.67 V (Figure 11c). As shown in Appendix A, the NG-CoFe/Mo_2_C-800 catalyst almost maintains its original morphology after the stability tests of ORR and OER. In Figure 11d, the ORR and OER activities of the NG-CoFe/Mo_2_C-800 are compared with those of bifunctional electrocatalysts in the literature, which further illustrates that NG-CoFe/Mo_2_C-800 can be considered an efficient ORR/OER catalyst. 

The ORR and OER mechanisms on NG-CoFe/Mo_2_C-800 in alkaline media are speculated in Figure 12. For ORR, the synergistic interactions among FeCo and Mo_2_C are mainly responsible for the high activity [75,76]. Among them, the Mo_2_C particles provide abundant active sites for ORR [38], and the ultrathin N-doped carbon layer coated on the alloy surface enhances the electron transfer efficiency. According to previous reports, the introduction of Mo_2_C can significantly reduce the dissociation potential of O_2_* and accelerate the ORR process [38,77]. Moreover, the interaction between Mo_2_C and CoFe provides a large contribution for facilitating the formation of intermediates (O* and *OH), thus enhancing the ORR activity of NG-CoFe/Mo_2_C-800 [42,78,79]. For OER, the structural interaction between CoFe and Mo_2_C should lead to the charge rearrangement, which helps to create more active sites (with active CoOOH/FeOOH species) on CoFe alloys and facilitates the smooth electron transfer process, thus enhancing the electrocatalytic activity [79,80]. Moreover, the porous structure of NG-CoFe/Mo_2_C-800 allows the electrolyte to make full contact with the catalyst, improving the mass transfer efficiency for ORR/OER [74,81]. Meanwhile, the graphitic carbon layer with excellent electrical conductivity protects the internal alloys/Mo_2_C from corrosion in harsh conditions, improving the ORR/OER catalytic stability [82,83,84,85].

## 4. Conclusions

In summary, by using CoFePBA/PVP as a precursor, we successfully synthesize a composite electrocatalyst (NG-CoFe/Mo_2_C-800) with CoFe alloy and Mo_2_C nanoparticles encapsulated in an N-doped carbon skeleton to achieve a promising bifunctional (ORR/OER) activity (ΔE = 0.67 V) in alkaline media. After 30,000 s of CA test (ORR), the current density of NG-CoFe/Mo_2_C-800 decreases by only 15.26% due to the presence of highly active metal species (Mo_2_C and CoFe alloy) protected by N-doped carbon, and after 1000 LSV cycles (OER), approximately 91.8% of the current density is maintained due to the in-situ generation of hydroxyl oxides (CoOOH/FeOOH) originated from the CoFe alloys. Mo_2_C doping plays an important role in modulating the carbon skeleton and FeCo alloy to form a unique CoFe/Mo_2_C-carbon encapsulated structure, which enhances the corrosion resistance of the catalyst (NG-CoFe/Mo_2_C-800) in harsh environments and maintains structural stability during the long-term ORR/OER cycles. This study provides a new strategy for the synthesis of carbon-coated metallic species, facilitating the development of efficient alloy/TMC-based electrocatalysts for bifunctional oxygen electrocatalysis.

## Data Availability

Data can be made available upon request.

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
