# Peer review of "CoFe Alloy-Coupled Mo2C Wrapped by Nitrogen-Doped Carbon as Highly Active Electrocatalysts for Oxygen Reduction/Evolution Reactions"

_nanomaterials, 2023, doi:10.3390/nano13030543_

Round 1
Reviewer 1 Report
The authors reported the synthesis of NG-CoFe/Mo2C composites and observed ORR/OER bifunctional activity in alkaline media. This was ascribed to the synergistic interactions between the structural components, where Mo2C was responsible for ORR and CoOOH and FeOOH were for OER. The paper can be publishable in the journal pending the following issues
1. On multiple occasions, the authors argued that the ORR/OER activity was due to the electronic interactions between the CoFe, Mo2C, and N-doped carbon. However, there is no experimental evidence to support such interactions. The only data included that can imply such interactions is the TEM image (Fig 5c). However, the TEM image only shows their physical distributions and cannot confirm if there is any electronic interaction (by the way, much better quality TEM Images are needed). Further structural characterizations are needed, in particular, XPS, XAS, etc.
2. Fig 1b, all samples show a broad peak around 25 degree. This is the carbon peak, and consistent with the Raman results. The authors need to revise the discussion.
3. The fact that CoOOH and FeOOH were produced in OER suggests that the carbon layers should be porous. This needs to be clarified, otherwise it will be difficult to understand how the core will decide the electrocatalytic activity.
4. Both Co and Fe consist of a large fraction of Fe2+ and Co2+ (Fig 3). Yet their contributions to the ORR/OER activity were not mentioned. This needs to be discussed in the text.
Reviewer 2 Report
This manuscript describes the CoFe alloyed-coupled Mo2C on N-doped carbon for OER/ORR catalysts. Based on the comprehensive works in this manuscript, I would like to support the publication of this manuscript in Nanomaterials after addressing the following points.
1) The authors are encouraged to provide the XRD patterns for the spent catalysts after ORR stability tests. (either in-situ or ex-situ)
2) XPS data points should be changed for better readability in Figures S1-S4 and Figure 3.
3) Explain the origin for the better wettability of NF-CoFe/Mo2C-800.
4) How about the electrochemical active surface areas for the given catalysts? (ref; doi: 10.20964/2016.06.71)
5) The author may want to cite the following recent works as they represent the significant improvements in the oxygen-based electrocatalysis field.
- 10.1016/j.cclet.2021.10.002 (CoFe-LDH)
- 10.1016/j.cej.2021.132174 (S/N co-doped Co/Fe)
- 10.1021/ja406242y (CoFe Prussian blue)
- 10.1016/j.chemphys.2005.05.038 (Theoretical investigation on (oxidized) metal surfaces)
- 10.1016/j.jcat.2020.12.002 (Ni-/Co-/Fe-/Cu-based Prussian blue analogues)
Reviewer 3 Report
This work reports a bifunctional material of NG-CoFe/Mo2C (800 °C) that can serve as OER and ORR catalysts. This catalyst exhibited superior multi-electrochemical performance. Overall, the experiments are very interesting and systematically studied, which deserves its publication in this journal. Therefore, I would like to recommend its acceptance in this journal after addressing the following points.
1. The synergistic interaction between Mo2C and CoFe alloy can generate many active sites for ORR/OER. The authors are suggested to provide the theoretical mechanism to confirm the synergistic effect between Mo2C and CoFe alloy.
2. The authors claimed that the inclusion of N enhances only the electrochemical ORR performance. It means there is no any role of N for OER and Why?
3. The presence of graphitized carbon not only effectively facilitates the charge transfer
among Mo2C, CoFe alloy, and carbon during electrocatalysis, but also reduces the
corrosion of Mo2C and CoFe alloy by the electrolyte and improves the stability of the
catalyst. The authors are suggested to provide an additional experiment to confirm the reduction of corrosion in NG-CoFe/Mo2C.
4. The authors are suggested to provide the TEM-EDS elemental mapping of the respective elements to confirm the successful synthesis of NG-CoFe/Mo2C.
5. On page no. 9, the authors claimed that “the porous structure should facilitate the mass diffusion via the internal pore channels, thus accelerating the ORR/OER kinetics.” Please provide a satisfactory reason.
6. On page no. 10, there is a typo error in writing the formula of Mo2C.
7. The RCT values during ORR and OER electrochemical characterization are the same, why?
8. Similar research on the electrocatalyst can be cited in the appropriate positions for the reference of data presentation and explanation. Journal of Colloid and Interface Science, Volume 618, 15 July 2022, Pages 475-482; Carbon, Volume 179, July 2021, Pages 89-99; Composites Part B: Engineering, Volume 239, 15 June 2022, 109992.
Round 2
Reviewer 1 Report
the revision is good.
Author Response
the revision is good.
RE: Thank you very much.
Reviewer 3 Report
All the required revisions are done.
Author Response
All the required revisions are done.
RE: Thank you very much.